# Tutorial: Guidelines for Single-Cell RT-qPCR

**DOI:** 10.3390/cells10102607

**Published:** 2021-09-30

**Authors:** Daniel Zucha, Mikael Kubista, Lukas Valihrach

**Affiliations:** 1Laboratory of Gene Expression, Institute of Biotechnology CAS, 252 50 Vestec, Czech Republic; daniel.zucha@ibt.cas.cz (D.Z.); mikael.kubista@ibt.cas.cz (M.K.); 2Department of Informatics and Chemistry, Faculty of Chemical Technology, University of Chemistry and Technology, 166 28 Prague, Czech Republic; 3TATAA Biocenter AB, 411 03 Gothenburg, Sweden

**Keywords:** single cell, sample collection, reverse transcription, preamplification, quantitative PCR, gene expression, RT-qPCR

## Abstract

Reverse transcription quantitative PCR (RT-qPCR) has delivered significant insights in understanding the gene expression landscape. Thanks to its precision, sensitivity, flexibility, and cost effectiveness, RT-qPCR has also found utility in advanced single-cell analysis. Single-cell RT-qPCR now represents a well-established method, suitable for an efficient screening prior to single-cell RNA sequencing (scRNA-Seq) experiments, or, oppositely, for validation of hypotheses formulated from high-throughput approaches. Here, we aim to provide a comprehensive summary of the scRT-qPCR method by discussing the limitations of single-cell collection methods, describing the importance of reverse transcription, providing recommendations for the preamplification and primer design, and summarizing essential data processing steps. With the detailed protocol attached in the appendix, this tutorial provides a set of guidelines that allow any researcher to perform scRT-qPCR measurements of the highest standard.

## 1. Introduction

Gene expression profiling has accelerated our knowledge on the progression of many diseases and injuries [1,2,3,4,5,6,7,8]. Originally, gene expression was studied on tissue autopsies, providing sufficient amounts of starting material for most applications. The incentive to study disease, development, or healing with greater detail shifted the direction toward single-cell profiling, which allows: (i) identifying and/or discovering novel cell subtypes [4,5,9]; (ii) understanding cell trajectories undertaken upon stimuli [10,11,12]; (iii) describing the heterogeneity of tissue composition [13,14,15]; and (iv) uncovering the cell type-specific interactions within complex tissues [11,16].

Traditionally, single-cell reverse transcription quantitative PCR (scRT-qPCR) represented the first method of choice for conducting single-cell gene expression measurements. In the last decade, its position was replaced by single-cell RNA sequencing (scRNA-Seq) that allows measurement of the complete transcriptome in thousands of cells in a single experiment. Despite this trend, scRT-qPCR still retains a strong position in the field thanks to its precision, sensitivity, wide dynamic range, and ease of use [17,18,19,20,21,22,23,24,25].

Although there have been several attempts to summarize the scRT-qPCR workflow, these are either outdated publications, not reflecting the recent development in the field, or they provide only a theoretical background, or pay attention to a specific part of the workflow [22,23,26,27,28]. Here, we aim to extend the discussion about the individual aspects of single-cell experiments [29,30], summarizing recent updates in the field and providing a practical guideline for scRT-qPCR measurement, translating the theory into hands-on practice. Collectively, we reviewed individual aspects of single-cell collection and material handling, reverse transcription (RT), preamplification (preAMP), quantitative PCR (qPCR), and data analysis (Figure 1). The Appendix A encloses a detailed exemplary protocol serving as a starting point for researchers in conducting their own scRT-qPCR measurements (Appendix A).

## 2. Sample collection

### 2.1. Preparation of Single-Cell Suspension

The scRT-qPCR workflow begins with the preparation of a single-cell suspension followed by the collection of single cells (Figure 2). While the former is simple for cell cultures or biofluids, dissociation of tissues may pose a serious challenge. A literature survey and on-site optimization are recommended to avoid common issues associated with inadequate protocols for preparation of single-cell suspensions, i.e., low yield, low viability, loss of vulnerable cell types, or activation of immediate-early genes [31,32,33]. The yield and viability are routinely examined using counting chambers or automated cell counters in combination with adequate staining (trypan blue, propidium iodide). The losses of vulnerable cell types may be controlled by inspection of antibody-labeled cells in the microscope or an RT-qPCR measurement of cell type marker gene expression before and after the dissociation procedure [34]. Lastly, the activation of stress-induced genes is minimized by low temperatures during dissociation and/or the use of transcriptional inhibitors [35,36]. As the use of low temperatures may hamper the dissociation efficiency, the application of psychrophilic proteases has been recently suggested [37].

### 2.2. Single-Cell Collection

Compared to scRNA-Seq, scRT-qPCR faces the challenge of a relatively low throughput. Unguided collection of cells from a heterogenous tissue is therefore non-effective, as targeted cell types may be collected only in a few cases per experiment. Therefore, cell populations of interest are typically identified by labeling delivered via gene constructs or antibodies conjugated with a fluorescence signal. The most established methods for collection of single-cell material are fluorescence-activated cell sorting (FACS), micromanipulation, and laser capture microdissection (LCM), although a few modern alternatives have been recently introduced as well [38]. The first two methods retrieve live single cells, whereas the last method usually requires material fixation. Minimizing the collection time reduces the risk of artificial alteration of the gene expression landscape [31]. Out of the three methods, only FACS can be looked upon as a quick, high-throughput method. In comparison, micromanipulation and LCM are laborious and time-consuming techniques, but they allow for visual inspection of the material. In addition, LCM retains the information on the spatial composition of the collected material, providing the much-needed tissue context [39]. For a detailed overview of collection methods, we refer the readers to several recent comprehensive reviews [38,40,41,42,43].

### 2.3. RNA Extraction and DNase Treatment

RNA extraction is not recommended for single-cell material because of the limited RNA concentration [44]. Cells are rather collected directly into lysis buffers, where they burst in the hypotonic environment and release their intracellular content. At this stage, RNases are not a principal threat to the released RNA. The more numerous extracellular RNases are washed away during preparation of the cell suspension and during cell collection. Intracellular RNases are, on the other hand, quickly diluted in the lysis buffer volume and/or inactivated by the lysis buffer components and low working temperatures [45]. Similar to RNA extraction, DNase treatment is not recommended as it increases the sample volume and dilutes the RNA. Amplification of the genomic DNA (gDNA) is prevented in the design of qPCR primers, which target exons separated by intron(s) of a substantial length (Section 5.1 Assay Design) [27,44].

### 2.4. Direct Lysis

Cells are collected directly into the lysis buffer. Besides rupturing the cellular membrane, the lysis buffer also acts as a stabilizing agent, protecting RNA from degradation and its adherence to plastic walls. Although the literature and companies offer multiple complex candidates, a simple solution of 0.1% BSA in nuclease-free water (NFW) maintains a high RNA quality even for extended storage time periods at room temperature (up to four hours) or repeated freezing and thawing [45]. An alternative approach is to collect cells directly into a reverse transcription buffer. Reverse transcription buffers should be, however, used with caution in terms of lysis and storage efficiency, as the performance will depend on the particular kit used.

### 2.5. Storage Conditions

Suitable capture and storage units for collected single cells are 96- or 384-well RNase- and DNase-free plastic plates. Not only can they be readily used in the next steps, but the processing of the entire plate also provides statistically adequate cell numbers for data analysis. The collected single-cell material should be stored at −80 °C. The plates need to be sealed properly with hardback foils manufactured to withstand low temperatures. Although occasional freezing and thawing of single cells should not hamper the sample quality for scRT-qPCR analysis, we raise caution against this practice [46,47].

### 2.6. Small Bulk Analysis

Single cells do not necessarily need to be sampled individually per collection vessel but may also be collected in small bulks numbering tens to hundreds of cells. Although small bulk analysis does not provide a true single-cell resolution, it increases the RNA concentration and allows the detection of transcripts that are present in just a few copies per cell (e.g., lncRNAs, transcription factors) [28,48,49]. The limiting factor of small bulk analysis is the maximum number of cells per lysis buffer volume ensuring proper cell lysis and dilutions of intracellular RNases as well as other potential inhibitors [45]. The maximal recommended number of cells collected per collection vessel depends on the cell type, volume, and properties of the lysis buffer as well as the volume of the co-transferred medium. A typical cell-to-volume ratio ensuring good results is ~100 cells per microliter of lysis buffer [45].

## 3. Reverse Transcription

Reverse transcription (RT) is an enzymatic reaction where RNAs are reverse transcribed into complementary DNA (cDNA) sequences (Figure 3). RNA molecules that fail to be transcribed are omitted from downstream processing steps and are not detected. This makes RT the primary bottleneck of the entire RT-qPCR workflow.

### 3.1. RT-Associated Variables

RT is often referenced as the least reliable step of the scRT-qPCR workflow [20,22,50,51,52,53,54]. This is attributable to multiple options for how RT can be performed. RT performance is dependent on the selection of: (i) the reverse transcriptase (RTase) [51,52,55,56,57]; (ii) primers [27,53,55,58,59]; (iii) additives [55,60,61]; and (iv) temperature profile [56,62,63,64,65,66]. Additionally, minute amounts of starting material (tens of picograms of RNA per single cell) add a layer of complexity related to the stochasticity of the quantification of small copy numbers [28,51,54,55,67].

### 3.2. RTase Selection

Key RTase parameters are absolute sensitivity and reproducibility, which are closely tied to the reaction efficiency. These and other parameters have been recently examined in a broad spectrum of RTases by two independent studies [56,61]. These studies showed that Maxima H- and SuperScript IV (both ThermoFisher) are currently the most efficient RTases, due to which they are recommended for single-cell applications. Moreover, both RTases are applicable not only in scRT-qPCR but also in scRNA-Seq thanks to their template switching activity. Their performance is supported by their high processivity, increased thermostability, high synthesis rate, robustness to inhibitors, and RNase H and strand displacement activity. High processivity enables reverse transcription of long transcripts (declared up to 20 kb). Increased thermal conditions loosen secondary structures, a known impairment to cDNA synthesis [63,65,68,69]. The higher synthesis rate shortens the experimental time and increases the RTases’ turnover. Insensitivity to inhibitors allows for the introduction of additives, broadening the RTases’ applicability. Strand displacement activity in combination with the lack of RNase H enables the synthesis of multiple cDNA molecules using a single RNA molecule as a template [56]. Template switching activity is a central property in several scRNA-Seq protocols, allowing the incorporation of additional sequences to the cDNA ends, which serve as barcodes and/or further cDNA amplification [68,70].

### 3.3. Priming Strategy

Although RTases represent a key reaction component, RT cannot be efficiently initiated without a primer. Priming also has an essential role in the reaction efficiency and specificity. In practice, two priming methods are used in scRT-qPCR: (i) 3′end-oriented oligo(dT), or (ii) random hexamers having the potential to prime the reaction from any site on the RNA. Oligo(dT) priming is selective for polyadenylated transcripts. However, this selectivity comes with the risk of 3′end bias, as the RTase may stall due to secondary structures or bound proteins preventing transcription of the 5′ end [53,65,68]. This should be, therefore, considered in the qPCR primer design. Random hexamers lack the selectivity for RNA types, are less sensitive to RNA quality (they do not rely on the presence of a polyA tail which is first affected by degradation), and can prime single RNA molecules from multiple sites [53,56,57]. To achieve a balanced transcript coverage, the two priming methods are often combined for superior performance. Combinations of both priming strategies with high-end performing RTases can deliver RT yields of over 100% [55,56,71]. Finally, increased primer concentrations were shown to increase reaction yields in bulk samples [58,59]; however, this effect was not studied for single-cell material.

### 3.4. RT Additives

Various compounds have been shown to increase the RT efficiency, e.g., tRNA [53], total extracted RNA [58,60], MgCl_2_, and betaine or trehalose [72]. Surprisingly, recent reports failed to reproduce their enhancing properties, questioning the further application of these additives [55,61]. Alternatively, physical separation of the reaction into smaller reaction chambers has been shown to increase the reaction efficiency [73]. Locally increased reagent concentrations and faster assembly of molecular machineries accelerate the rate of interactions, which results in an improved RT efficiency. As this approach is hardly applicable on the already small volumes of single-cell measurements, the molecular crowding effect may be delivered by the addition of polyethylene glycol (PEG 8000), a mechanism already implemented in recent scRNA-Seq protocols [61,74].

### 3.5. Temperature Profile

The majority of RT protocols consist of two steps. Firstly, mixtures of dNTPs, spike-ins, and RT primers are added to the sample and incubated at an elevated temperature. The high temperature melts the secondary structures and ensures that the primers can anneal more equally along the template length. Therefore, it is not recommended to omit this step as it may negatively affect the reaction efficiency [56]. In the second step, an RT buffer, RNase inhibitor, and RTase are added. The use of RNase inhibitors is recommended to minimize the risk of lower reaction yields due to the activity of the contaminating RNases.

### 3.6. Setting RT Reaction

Good laboratory practices represent a key aspect to deliver reproducible and reliable results. Before RT mastermixes are prepared, all reagents need to be vortexed and spun down, except for protein-based reagents (RNase inhibitors and RTases). Protein-based reagents are only briefly spun down, and, except for during the pipetting time, they are best kept at the recommended storage conditions. dNTPs and primers are typically prepared in aliquots that minimize repeated freeze–thawing and are supplied in concentrations ready for use. Once prepared, the RT mastermix is added directly to the sample, and by doing so, one avoids the unnecessary transfer of single-cell material. Minimal RT volumes are preferred as they maintain high concentrations of the target molecules and reagents, improving the outcome of scRT-qPCR [44,51,61,68,73,75]. The use of RNase-free plastic is a necessity [65]. Biological replicates are preferred over technical replicates [22,23]. cDNA is stored at 4 °C for up to 24 h, but for long-term storage, −20 °C is preferable.

### 3.7. Quality Control

RNA spike-ins represent an effective tool for quality control throughout the entire RT-qPCR workflow [76]. RNA spike-ins are molecules of an artificial and unique sequence that are added in equal amount into each RT reaction and consequently measured by a dedicated qPCR assay. A uniform signal across all the samples shows that none of the samples were subjected to pipetting error or inhibition. Commercial spike-in kits differ in the complexity of their contents. Simple options contain a single RNA transcript variant (e.g., TATAA RNA Spike-in, TATAA Biocenter), whereas more complex alternatives are composed of a panel of RNA transcripts varying in length and abundance (e.g., ERCC Spike-in, ThermoFisher), mimicking the transcriptome complexity [77]. The latter option is more suitable for applications such as scRNA-Seq, where the total number of screened assays is not a limiting factor. Being a quality control reagent, it is important to prevent a spike-in’s repeated freeze–thawing; thus, storing it aliquoted, similar to dNTPs and primers, is advised. As a rule of thumb, the measured Cq signal for a spike-in assay should not deviate by more than one cycle from the mean Cq spike-in signal per plate.

## 4. Preamplification

Single-cell RT is performed in low volumes to prevent unnecessary RNA dilution and loss of sensitivity [22]. This implies that only a limited sample volume is available for follow-up qPCR measurements. Quantification of multiple targets per cell therefore requires their enrichment by preamplification (Figure 4). Noteworthy, preamplification (preAMP) is not mandatory for targeted profiling of a limited number of transcripts where sample dilution is not applied.

### 4.1. Targeted vs. Global Preamplification

Targeted preAMP is a regular multiplex PCR, but instead of a single pair of forward and reverse primers (further referred to as assay), a multiplicity (tens) of assays are employed in the reaction at once. Typically, assays used in targeted preAMP are identical to those used in the later qPCR step. The pool of assays is selected to address the experimental question of interest, enriching the reaction contents exclusively for cDNAs of targeted transcripts [78,79]. cDNA transcripts that are not amplified by the primer assay pool cannot be quantified in subsequent qPCR as they are diluted to a non-detectable concentration in the further steps. An alternative to target-specific preAMP is global preAMP, where a single pair of primers is used to amplify cDNA copies globally for all transcripts [72,80]. The flexibility of global preAMP is, however, paid off in a reduced yield and sensitivity and increased variability [80]. Due to these reasons, global preAMP is not frequently used in scRT-qPCR experiments.

### 4.2. Reaction Parameters

Well-optimized preAMP is a result of several fine-tuned parameters including: (i) individual assay efficiency; (ii) number of assays per preAMP assay pool; (iii) concentration of the preAMP assay pool in the reaction; (iv) reaction volume; (v) temperature profile; and (vi) the number of amplification cycles. Well-optimized preAMP aims to deliver a reproducible output for each assay [23]. An assay employed in scRT-qPCR needs to be target-specific, produce no primer dimers, not amplify gDNA, and showcase a reproducible efficiency of >90% (see Section 5 Quantitative PCR for further details) [22,27,81]. Although efficient in separate assays, the pool of preAMP assays has potential to give rise to unspecific products. Still, more complex preAMP assay pools (>50 assays) produce fewer artefacts than those combining less assays, deliver higher yields, and are less variable [79]. This is attributed to the increased number of interactions that make the formation of stable primer dimers less likely. The formation of unspecific products is also dependent on the primer concentration and number of cycles. To minimize their production, the recommended number of cycles is between 18 and 22 cycles, and the final primer concentration should be around 40 nM, which is about ten times less than in standard single-plex PCR [79]. Lastly, to account for annealing heterogeneity of the preAMP assay pool, the annealing temperature is set as the average of the respective assays (optimally within ± 1 °C of the assays’ annealing optima), and the annealing time is prolonged to three minutes [79].

### 4.3. Validation

Before any preAMP assay pool is employed, it is necessary to inspect whether its individual assay components maintain a high level of performance even when combined in the assay pool. Screening of more assays provides a more encompassing overview, but screening of the entire assay list would also prove to be very laborious and an expensive task. In our practice, a subset of assays targeting cell type markers, genes of interest, and spike-ins (minimum of eight in total) is sufficient for general assessment of the preAMP performance. Validation is performed on random cDNA, gDNA, and non-template control (NTC) samples by inspecting the cycle of quantification (Cq) for non-preamplified and preamplified samples [81]. The first control is focused on the variability of deltaCq (SD_∆Cq_), where deltaCq is the difference in Cq between non-preamplified and preamplified cDNA. The decision on the degree of tolerated variability is left to the researcher; however, its value should approach the standard qPCR variability (SD_Cq_) of ~ 0.2 among replicates [22]. In our perspective, a preAMP assay pool with assays scoring SD_∆Cq_ ≤ 0.5 is considered robust and reliable. A step-by-step guide on how to calculate SD_∆Cq_ can be found in the Appendix A. Secondly, the measurement of gDNA provides insight into the sensitivity of each assay to the gDNA background before and after preAMP. Thirdly, the NTC sample reports on the production of unspecific products and serves for the identification of assays that may report false positive signals in preamplified samples.

### 4.4. Setting preAMP Reaction

After validation of the preAMP reaction, the finalized reaction setup begins with the selection of assays of interest. Apart from the assays determined in the experimental design, the assay list ought to include control assays as well, e.g., a spike-in assay or an assay to control the gDNA content [82]. The preAMP assay pool is prepared on ice and stored in aliquots to avoid primer degradation during freeze–thaw cycles. Any PCR mastermix can be used, although those containing more enzymes are preferred. The selected PCR mastermix does not need to include fluorescence reporters, as the reaction is not visualized by a standard. Undiluted cDNA samples are typically mixed with a mastermix in a 1-to-9 ratio. Here, 10× cDNA dilution is mandatory to prevent PCR inhibition by RT reagents [54,58,83,84]. Although it is theoretically possible to process an entire cDNA sample and thus achieve maximal sensitivity, for cost reasons, only a portion of single-cell cDNA is routinely analyzed. This consequently determines the portion of the single-cell transcriptome analyzed in the RT-qPCR workflow. After the preAMP reaction, the preamplified material needs to be immediately diluted and frozen at −20 °C to terminate polymerase activity and production of potential artefacts.

## 5. Quantitative PCR

Quantitative PCR (qPCR) is the third and final laboratory step of the scRT-qPCR workflow (Figure 5). It extends the previous preAMP step, as the assays are usually shared for the preAMP and qPCR steps. Contrary to preAMP, the qPCR performance is visualized after every cycle and is typically run in simplex (single assay per reaction) using non-specific dyes such as SybrGreen or EvaGreen.

### 5.1. Assay Design

The quality of each qPCR assay is determined by its design [25,27,65,81]. The basic rules for efficient primer design can be summarized in a few characteristics: (i) primer design over introns (avoiding gDNA co-amplification); (ii) target 3′end proximal exons (reducing 3′end oligo(dT) priming bias); (iii) primer length of an average of 20 bases (±5 bases); (iv) symmetric (similar in size); (v) amplicons of 70–120 bases; (vi) balanced GC content (40–60%); (vii) sequence free of consecutive nucleotide repetition (e.g., no ‘GGGGG’); (viii) no intra- or inter-primer 3′end complementarity (reducing risk of primer dimers); and (ix) no hairpin structures. An efficient method of finding specific primers is offered by PrimerBlast [85]. Detailed characterization of primers may be further performed by NetPrimer by PREMIER Biosoft or OligoAnalyzer Tool by Integrated DNA Technologies. For a more in-depth view on primer design, see Bustin and Huggett [25]. When preparing new assays, designing two or three variants per target at once makes the follow-up validation process more time-efficient.

### 5.2. Assay Validation

The performance of each new assay needs to be validated experimentally. Typically, a set of cDNA, gDNA, and no-template samples is used for this purpose. A well-performing assay needs to report: (i) a reproducible signal for the cDNA samples (SD_Cq_ < 0.2); (ii) no-to-minimal gDNA signal; and (iii) no signal for no-template samples. Additionally, post-PCR melt curve analysis is applied to inspect the reaction specificity in the reactions using non-specific dyes. A singular peak suggests the presence of a specific product, whereas distortion of the peak shape or the presence of another peak signalizes co-amplification of unspecific products [18,22,25]. Complementarily, separating the qPCR products on gel electrophoresis highlights the amplicon composition. Next, the assay efficiency is determined by standard curve analysis [24]. PCR products of the cDNA samples are frequently used as a template for ten-fold dilution series across at least six orders of magnitude (e.g., 10× to 10^6^×). Templates are prepared individually for each assay, and the standard curves are measured in replicates. The assay efficiency is calculated according to the formula
*E* = 10^−1/a^(1)
where *a* refers to the slope of the standard curve. Assays intended for single-cell application are recommended to record an efficiency of *E* ≥ 0.9 (90%) [23,28].

### 5.3. Limit of Detection and Limit of Quantification

Low target concentrations are an inherent problem in single-cell measurements [22,23,28,55]. To report changes of low-expressed transcripts among samples reliably, technical limitations of assays need to be determined. The limit of detection (LoD) relates to the lowest sample concentration that is reported in 95% of the cases. The limit of quantification (LoQ) determines the lowest concentration which can be reproducibly measured. The theory of reporting minimal copy numbers sets the LoD to three molecules per reaction, but in reality, this number increases up to ten molecules [28,67,86,87]. Reporting limited copy numbers is also obscured by the low reproducibility of measurements, observed as increased Cq variation (SD_Cq_) [22,56]. The LoQ reflects this uncertainty, and it is up to researcher to consider whether the assay’s performance is sufficient for the intended application. Illustratively, the variation in reporting ten molecules is approximated to SD_Cq_ ~0.5, which is attributed to the natural sampling error [22]. Similar to the efficiency, the LoD and LoQ are determined by standard curve analysis. Details on LoD–LoQ standard curve analysis and the calculation are listed in the Appendix A.

### 5.4. Setting the qPCR Reaction

qPCR is well known for its precision, and therefore it is preferred to use biological replicates over technical ones [20,22,23]. When the goal is to analyze a few genes in tens of cells, conventional 384-well qPCR is most likely the suitable format. If the throughput of conventional qPCR is too small to encompass the experimental design, high-throughput platforms are designed to meet the requirements of simultaneous profiling of tens of cells with tens of assays (e.g., BioMark, Fluidigm). The qPCR precision also has potential to uncover unwanted pipetting imprecision as well as cross-contamination. Accumulation of pipetting mishaps can be reduced by a clever pipetting plan that accommodates pipetting with an eight-channel pipette. Preparation of mastermixes follows the same recommendations as in the previous RT-qPCR steps.

### 5.5. Quality Control

As with the previous steps, proper quality controls are essential to ensure a high quality of reported data. Therefore, each sample list should include an NTC, positive control, gDNA sample, and interplate calibrator (IPC). The IPC may serve a bulk sample having a similar representation of transcripts to that analyzed in single cells. The IPC then serves as the positive control and IPC at the same time. The IPC is aliquoted to avoid repeated freeze–thawing and is measured by a reliable assay on each plate. As the IPC is shared across the plates, the respective Cq values can be accordingly re-scaled to minimize technical differences between individual plates [54,87]. The NTC and gDNA samples control for assay specificity. In the list of assays, assays specific to genomic content (e.g., ValidPrime [82]) and spike-in assays should not be missing. Before initiating a large experiment, pre-screening on the quality of collected single cells can be advantageous. Measuring several markers reflecting the cell type or cell state can be performed even on non-preamplified material.

## 6. Data Analysis

### 6.1. Single-Cell Data Specifications

Although single-cell data generally resemble those generated on bulk samples, some features are unique [28,87]. The main differences are: (i) increased frequency of false positive reactions due to unspecific products generated in the preAMP step; (ii) higher frequency of missing data attributed to the low amount of starting material, transcriptional bursting, and limited sensitivity of the scRT-qPCR workflow (false negatives); and (iii) larger data matrices requiring different types of analysis and visualization (Figure 6).

### 6.2. Data Pre-Processing

At the beginning, assay melting curves are inspected to determine the signal specificity. Co-amplification of gDNA or unspecific products is observable by the presence of multiple peaks or by a distorted shape of specific peaks [18,25]. Inspecting peaks of NTC and gDNA samples helps guide the analysis. Signals characterized by unreliable peaks are discarded. Further, off-scale data are identified and removed as they most likely represent technical artefacts as well. The off-scale values are defined by the workflow and the assay’s sensitivity (LoQ or LoD). For low-throughput measurements when samples are not preamplified, Cqs above 40 are typically discarded. In high-throughput measurements requiring preAMP, the cut-off of 28 cycles is routinely applied [87,88]. In the next step, the dataset containing only reliable Cqs is transformed to relative quantities (RQs). RQs are calculated for every assay separately according to the formula
(2)$$RQ=2^{Cq_{MAX}-Cq}$$
where *Cq_MAX_* is the highest measured *Cq* per assay in a given dataset. A substantial portion of the scRT-qPCR readout is composed of missing values. This is caused by the absence of the targeted molecule, or it is a consequence of a limited RT-qPCR sensitivity [28]. To reflect both possibilities, missing values are imputed. Imputation of values that are close to the lowest specific signal having RQ = 1 (imputation of, e.g., RQ = 0.5 or 0.25) makes the missing values less distinct from the measured values and rather reflects an insufficient reaction sensitivity. Conversely, imputation of extreme values (RQ = 0.125 or less) highlights the importance of the assays with a more distinct pattern between the studied treatments or cell types. Therefore, more extreme imputation highlights differences between the experimental conditions. Unwantedly, imputation can also ascribe significance to cells of a lower quality (i.e., due to poor handling), as they have more missing values than readily prepared cells. As a rule of thumb, cells should be considered for discarding if they report 50% more missing values than the average for the rest of their cell type. Finally, the entire dataset is log_2_-transformed which typically causes a log-normal distribution of the data and allows application of statistics methods based on this distribution, e.g., *t*-test and ANOVA [44,87,88].

### 6.3. Descriptive Analysis and Data Visualization

It is good practice to begin the exploration of any single-cell dataset by descriptive statistics. The number of positive reactions per assay, but also per cell, can help with identification of key genes defining cell populations or experimental conditions. If an assay of known expression is measured, inspection of its signal may serve as a good quality control. Next, dendrograms and heatmaps are other powerful approaches that facilitate the exploratory process [87]. Gene–gene correlation plots or histograms of biologically relevant assays may provide insights into ongoing biological processes. Regular statistical tests as the *t*-test or ANOVA find use in assessing the significance of differential expression, keeping in mind the correction for multiple testing, i.e., Bonferroni corrections and others [87]. Finally, dimensionality reduction tools such as principal component analysis (PCA) or self-organizing maps (SOM) are powerful tools to visualize the data, and to identify clusters of cells with similar properties, and genes responsible for the separation [28,87,88,89]. As soon as clusters are identified, they can be iteratively analyzed by the descriptive analysis approaches described above. Dimensionality reduction methods often require data scaling that equalizes the statistical power across assays. Mean centering and z-scoring are the most frequently used approaches for scRT-qPCR datasets. For a step-by-step guide on scRT-qPCR data analysis and clustering, see [87,88].

## 7. Conclusions

The field of single-cell gene expression analysis is characterized by constant development of new technologies and data analysis tools. In this vastly evolving field, scRT-qPCR represents a well-established and standardized method whose performance has been proved by hundreds of studies. Although the method may be implemented by any laboratory having the basic equipment, allowing bulk expression analysis, single-cell application possesses specific challenges that need to be carefully considered. Here, we reviewed these challenges, provided theoretical recommendations, and supplemented them with practical step-by-step guidelines. We believe that our effort will facilitate implementation of the technique in many standard research laboratories, promote standardization, and improve the reporting of data. This will open up new avenues to address numerous biological questions that cannot be resolved with traditional bulk analysis.

## Figures and Tables

**Figure 1 cells-10-02607-f001:**
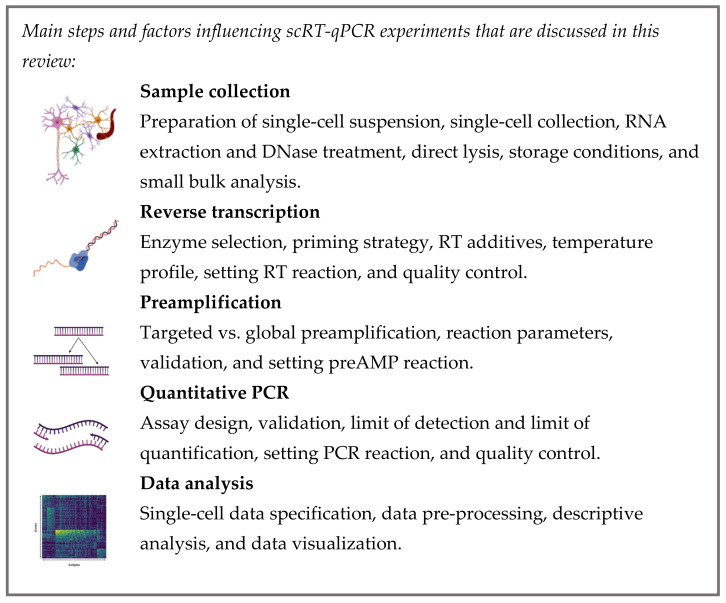
scRT-qPCR overview.

**Figure 2 cells-10-02607-f002:**
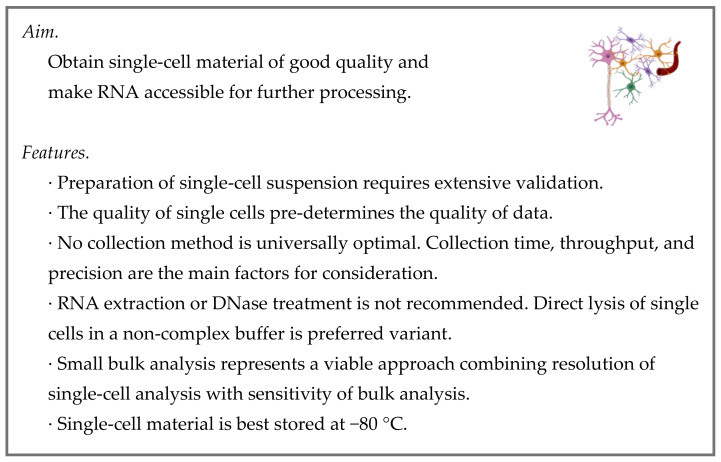
Sample collection.

**Figure 3 cells-10-02607-f003:**
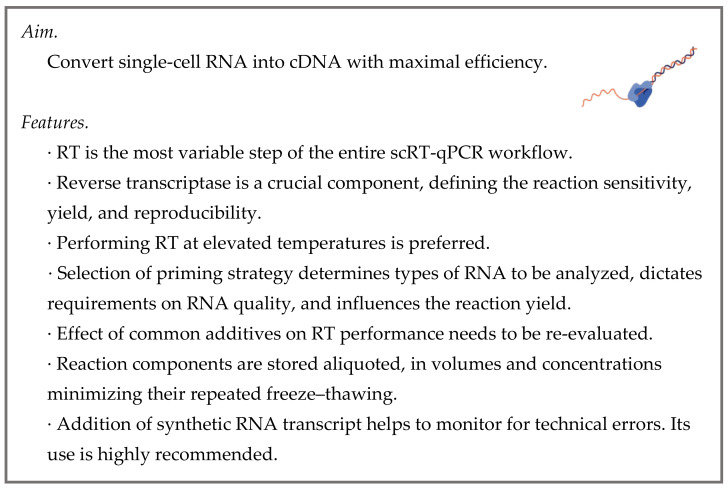
Reverse transcription.

**Figure 4 cells-10-02607-f004:**
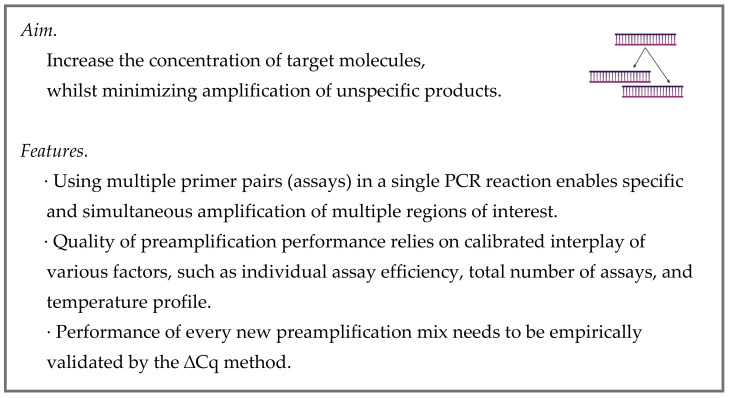
Preamplification.

**Figure 5 cells-10-02607-f005:**
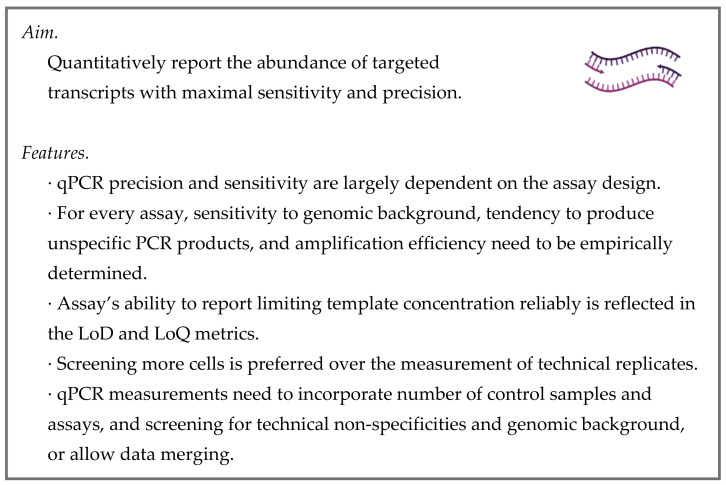
Quantitative PCR.

**Figure 6 cells-10-02607-f006:**
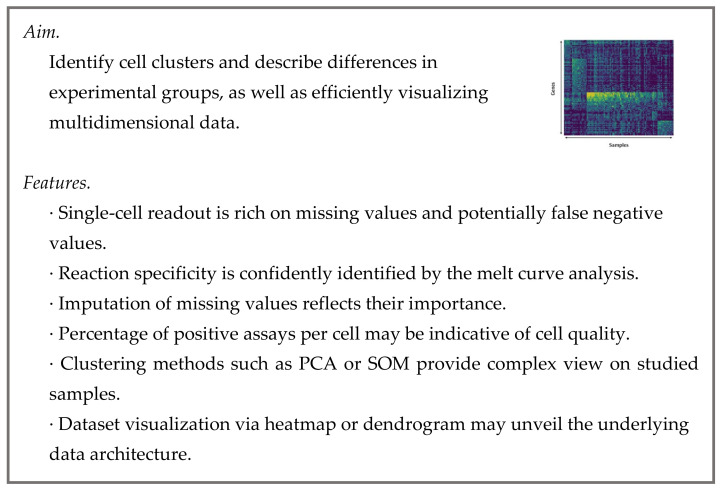
Data analysis.

## Data Availability

The data presented in this study are available on request from the corresponding author.

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
