# Peer review of "Tutorial: Guidelines for Single-Cell RT-qPCR"

_cells, 2021, doi:10.3390/cells10102607_

Round 1
Reviewer 1 Report
The manuscript by Zucha et al. summarizes and describes the steps of single cell RNA preparation for transcriptomic profiling with RT-qPCR. Considering the great interest in single cells profiling it is necessary to standardize or at least introduce “good practice” in the qPCR field so that high research standards are maintained. Such manuscript is really needed in the field and I am sure it will be well appreciated. In my opinion, the manuscript is well written and covers well the aspects of RT-qPCR for single cell analysis. Though some information is “hidden” in the supplementary data and could be more highlighted in the main manuscript. Additionally, can cDNA prepared according to such protocol be used in RNA-Seq?
I would be grateful for more explanation regarding the quality control of single cells material. The detailed questions are listed below.
- In the secion 3.3. Priming strategy the authors described most commonly used RT priming strategies – poliA and random hexamers. It would be worth mentioning what is the influence of the given priming strategy on the potential to amplify genomic DNA from single cells.
- In the section „3.7. Quality control.” – what should be the accepted variation in spike-in after the whole procedure to say that the sample was correctly processed? At what point sample should be discarded based on the lower spike-in level?
- The section “4.3. Validation.” Is not clearly described. Differences in Cqs of non-amplified and amplified sample is not the same as SD of delta Cqs. What about comparing the level of individual genes expression among each other to see if proportions of genes expression levels have not changed significantly after preamplification?
- Section “4.4. Setting preAMP reaction.” – maybe it should be again re-iterated that setting preAMP and selection of assay of interest is a decision based on the previous steps of the preamp validation, i.e. if one validated preamplification of 15 targets, these 15 targets should be used in the final protocol. Adding/removing targets at that point would not be matching previous steps of the validation.
- In the section “5.2. Assay validation.” I would also add running electrophoresis to check that designed primers amplify the fragment of specific length. It also makes it easier to see if there are any non-specific bands. Electrophoresis is mentioned in the supplementary protocol, but adding it to the main manuscript would indicate its importance.
- In the section “5.5. Quality control.” – shall gDNA contamination be tested on each plate or can it be skipped if the initial validation of the primer showed that it does not amplify gDNA?
- Section “6.2. Data pre-processing.” - In the given formula for RQ,how do the Authors define “CqMAX” - the highest measured Cq per assay. Measured when? Please specify.
- Section “6.2. Data pre-processing.” – poor quality of the input material (degradationof RNA) can also be a reason for missing qPCR values. What is the percentage of values that can be imputed and what is the relation to poor quality of the cells?
In the supplementary data:
- How the results of spike-in should be interpreted? What is the “allowed” variation? And when should cells be removed from the analysis due to low level of detected spike?
- The authors state that quality control of the samples (section 3.4) is optional. However including samples with poor quality for further transcriptomic profiling can also lead to biased results.
In the same paragraph the authors suggest to do quality check on non-pre amplified sc cDNA. How many reactions can be performed according to the protocol they describe? What about testing RNA spike as quality control?
More general:
- What about global preamplification of the transcriptome? Why focus only on targeting preamplification of cDNA?
- Maybe it is worth adding to what extend the described guidelines apply to other existing protocols for single cells RNA preparation for transcriptomic profiling. eg. dilution of cDNA can vary etc.
Author Response
We thank both reviewers for positive evaluation of our work.
#### Response to the MDPI Cells reviewer 1
Q: Can cDNA be prepared according to such protocol be used in RNA-Seq?
The here-presented review aims to introduce laboratory practices that have utility beyond single-cell RT-qPCR. Despite single-cell RT-qPCR and RNA-Seq in theory share multiple workflow steps, RNA-Seq protocols contain steps not necessary for RT-qPCR (end-repair, fragmentation, ligation of PCR handles, etc), and this diversion makes this protocol not directly applicable for sequencing applications. Still, cDNA prepared by polyA priming might be in theory used in scRNA-seq protocols, but only if certain modifications into oligodT primer are included. This involves addition of extra sequences for PCR amplification or in-vitro transcription at 5’ end of oligodT primer. These modifications are however not employed in standard scRT-qPCR protocols (including ours).
---
Q1: In the section 3.3. Priming strategy the authors described most commonly used RT priming strategies – poliA and random hexamers. It would be worth mentioning what is the influence of the given priming strategy on the potential to amplify genomic DNA from single cells.
A1: Amplification of genomic DNA under reverse transcription conditions is expected to be minimal. Firstly, the RT reaction temperature (with 42 - 65 °C range) is insufficient for complete melting of genomic DNA double strand. Next, the thawed regions would need to be efficiently primed, which is plausible with random hexamers, but very unlikely with oligodT primers. Primed gDNA regions would then need to be efficiently (reverse) transcribed, but as reverse transcriptase has high preference for RNA-DNA template, the efficiency of gDNA transcription is limited. In overall, amplification of rare single-cell transcripts is challenging even when targeted, as their small numbers are cause for unreliable quantification. Based on this we reason that the contribution of gDNA primed with RT primers is negligible in the overall single-cell transcription profile.
---
Q2: In the section „3.7. Quality control.” – what should be the accepted variation in spike-in after the whole procedure to say that the sample was correctly processed? At what point sample should be discarded based on the lower spike-in level?
A2: Since spike-in molecules are routinely added in quantities that are not subject to reaction sensitivity, we would recommend to treat discrepancies in the spike-in quantification with respect to RT-qPCR's technical limitations. Combined with our experience, samples are discarded when their spike-in Cq signal deviates as much as one cycle from the average spike-in Cq. This recommendation has been added into main text.
---
Q3: The section “4.3. Validation.” Is not clearly described. Differences in Cqs of non-amplified and amplified sample is not the same as SD of delta Cqs. What about comparing the level of individual genes expression among each other to see if proportions of genes expression levels have not changed significantly after preamplification?
A3: Thank you for pointing out insufficient naming convention of the validation unit. Certainly, one can also inspect the absolute change in the Cq value between the non-preamplified and the preamplified sample. However, single-cell RT-qPCR is not a tool of absolute quantification, but rather it is interpreted in relative quantities (see part 6.2 Data preprocessing). This implies that as along as the targeted transcript is invariably amplified in the preamplification reaction, the technical bias in the resulting Cqs is minimal. For this reason, we advise to inspect SD of delta Cq, as this variability could negatively affect reported results. To make this clear in the main text, reference for the supplementary material was added for precise calculation.
---
Q4: Section “4.4. Setting preAMP reaction.” – maybe it should be again re-iterated that setting preAMP and selection of assay of interest is a decision based on the previous steps of the preamp validation, i.e. if one validated preamplification of 15 targets, these 15 targets should be used in the final protocol. Adding/removing targets at that point would not be matching previous steps of the validation.
A4: It is indeed the case that during the preamplification validation all assays that appear in the finalized reaction setup ought to be tested. On the other side, in this validation step it is the variance of assay performance in the interaction with other assays in the pool that is in focus (because for the assay to be considered for preAMP, it is validated first as described in the 5.2 qPCR validation section). Meaning that if the components of the preAMP assay pool are exchanged in another round of experiments, again the tens of included assays ought to re-validated. This would turn out to be very expensive and time-consuming practice. Therefore in this step we only reflect that minimum of screened assays should be eight (supplementary material) and we clarified the goal of this step in the main text.
---
Q5: In the section “5.2. Assay validation.” I would also add running electrophoresis to check that designed primers amplify the fragment of specific length. It also makes it easier to see if there are any non-specific bands. Electrophoresis is mentioned in the supplementary protocol, but adding it to the main manuscript would indicate its importance.
A5: Thank you for making note of this. The sentence has been added to the main text.
---
Q6: In the section “5.5. Quality control.” – shall gDNA contamination be tested on each plate or can it be skipped if the initial validation of the primer showed that it does not amplify gDNA?
A6: Indeed, if an assay did not amplify gDNA in previous validation steps, gDNA can be dropped from the sample list. However, to maintain the good practice, we keep the necessity of gDNA samples as a control in the main text (in the case that assay validation was not performed in an optimal way, using different chemistry, measured in different instruments etc.). Moreover, in the case of high-throughput measurements in microfluidics devices, where all samples are combined with all assay, it is expected that some assays will amplify gDNA, therefore the addition of gDNA control sample is mandatory to distinguish true signal from gDNA background.
---
Q7: Section “6.2. Data pre-processing.” - In the given formula for RQ,how do the Authors define “CqMAX” - the highest measured Cq per assay. Measured when? Please specify
A7: By CqMAX it is meant the highest measured Cq per assay in a given dataset. The clarification was added in the main text.
---
Q8: Section “6.2. Data pre-processing.” – poor quality of the input material (degradation of RNA) can also be a reason for missing qPCR values. What is the percentage of values that can be imputed and what is the relation to poor quality of the cells?
A8: Very good question, thank you. Poor quality cells can be noticed most easily by substantially reduced expression of the marker genes for the cell type, or by having too many missing values across the measured assays (% of positive calls for the assay list). It is however impossible to pose strict limit on the change of marker expression or assay positivity, since the measured assay panel is experiment-specific. This means that for experimental setups where only rare transcripts are targeted, as opposed to pure 'marker panel', the increased rate of dropout events is to be expected. However, as rule of thumb, cell scoring just half of positive calls than the rest of measured cells (of the same cell type) on the same plate would be considered of poor quality and discarded. The clarification was added in the main text.
---
Q9: How the results of spike-in should be interpreted? What is the “allowed” variation? And when should cells be removed from the analysis due to low level of detected spike?
A9: Please, see our respond to question 2. Cells deviating as much as one Cq from the mean should be discarded.
---
Q10: The authors state that quality control of the samples (section 3.4) is optional. However including samples with poor quality for further transcriptomic profiling can also lead to biased results. In the same paragraph the authors suggest to do quality check on non-pre amplified sc cDNA. How many reactions can be performed according to the protocol they describe? What about testing RNA spike as quality control?
A10: The reviewer is certainly correct on how analysis of cells of poor quality can lead to biased results. Therefore, we suggest pre-testing as additional quality control tool. However, if a researcher is confident with the quality of collected cells, this step is not mandatory.
In our setup, reverse transcription is usually performed in 10 ul, 4 ul are needed for preAMP and 6 ul are left for quality control. With accounting for pipetting losses, this provides capacity to test up to 5 assays per single cell (using 1 ul of cDNA/reaction). Information on the maximal number of assays that can be measured was added to the supplementary text.
Measurement of RNA spike-in is a good advice, as it monitors any technical issue in the RT-qPCR workflow, e.g. reverse transcription/preAMP failure or pipetting error. However, RNA spike-in does not control for cell quality, therefore additional controls (incl. the pre-testing procedure) are recommended.
---
Q11: What about global preamplification of the transcriptome? Why focus only on targeting preamplification of cDNA?
A11: The flexibility of global preamplification comes at the price of poorer yield, sensitivity and higher variability, as highlighted by Kroneis et al (2017) and commented in the main text.
---
Q12: Maybe it is worth adding to what extend the described guidelines apply to other existing protocols for single cells RNA preparation for transcriptomic profiling. eg. dilution of cDNA can vary etc.
A12: Our goal with this review and its supplementary material was to provide the reader with a detailed protocol that can be immediately put to use. Certainly, adjustments to the protocol are feasible, and maybe even expected by the end-point user, as he is to gain confidence and expertise with our review being the well-defined starting point.
Reviewer 2 Report
Zucha and collaborators have written a manuscript that provides a set of guidelines for the single-cell RT-qPCR. It is an extensive tutorial in which they discuss limitations of single-cell collection methods, provide protocol recommendations for the different steps and summarize essential data processing steps. Supplementary material also encloses detailed protocol serving as a starting point for conducting own scRT-qPCR measurements. Furthermore, included images are very useful to summarize each step of the protocol. The article is well written and provides enough detail to allow any user to perform the single-cell RT-qPCR process. A possible suggestion would be to reduce bibliographic references since in some cases there are too many (i.e page 1 line 31 or page 9 line 158).
In my opinion, the manuscript could be accepted in the present form.
Author Response
We thank the reviewer for positive evaluation of our work.
The list of bibliographic references was shortened.